# Plasma Lipidomics Analysis Reveals the Potential Role of Lysophosphatidylcholines in Abdominal Aortic Aneurysm Progression and Formation

**DOI:** 10.3390/ijms241210253

**Published:** 2023-06-16

**Authors:** Ting Xie, Chuxiang Lei, Wei Song, Xunyao Wu, Jianqiang Wu, Fangyuan Li, Yanze Lv, Yuexin Chen, Bao Liu, Yuehong Zheng

**Affiliations:** 1Clinical Biobank, Department of Medical Research Center, Peking Union Medical College Hospital, Chinese Academy of Medical Sciences and Peking Union Medical College, Beijing 100730, China; xietingpumch@163.com (T.X.); doris_lfy@163.com (F.L.); 2Department of Vascular Surgery, State Key Laboratory of Complex Severe and Rare Diseases, Peking Union Medical College Hospital, Chinese Academy of Medical Sciences and Peking Union Medical College, Beijing 100730, Chinaliubao72@aliyun.com (B.L.); 3Department of Medical Research Center, State Key Laboratory of Complex Severe and Rare Disease, Peking Union Medical College Hospital, Chinese Academy of Medical Sciences and Peking Union Medical College, Beijing 100730, China; sw-yy1990@163.com (W.S.);

**Keywords:** lysoPC, abdominal aortic aneurysm, lipid metabolism, lipidomics, HDL-c

## Abstract

Abdominal aortic aneurysm (AAA) is hallmarked by irreversible dilation of the infrarenal aorta. Lipid deposition in the aortic wall and the potential importance of a lipid disorder in AAA etiology highlight the need to explore lipid variation during AAA development. This study aimed to systematically characterize the lipidomics associated with AAA size and progression. Plasma lipids from 106 subjects (36 non-AAA controls and 70 AAA patients) were comprehensively analyzed using untargeted lipidomics. An AAA animal model was established by embedding angiotensin-II pump in ApoE^-/-^ mice for four weeks and blood was collected at 0, 2 and 4 weeks for lipidomic analysis. Using a false-discovery rate (FDR) < 0.05, a group of lysophosphatidylcholines (lysoPCs) were specifically decreased in AAA patients and mice. LysoPCs were principally lower in the AAA patients with larger diameter (diameter > 50 mm) than those with a smaller size (30 mm < diameter < 50 mm), and levels of lysoPCs were also found to be decreased with modelling time and aneurysm formation in AAA mice. Correlation matrices between lipids and clinical characteristics identified that the positive correlation between lysoPCs and HDL-c was reduced and negative correlations between lysoPCs and CAD rate, lysoPCs and hsCRP were converted to positive correlations in AAA compared to control. Weakened positive correlations between plasma lysoPCs and circulating HDL-c in AAA suggested that HDL-lysoPCs may elicit instinctive physiological effects in AAA. This study provides evidence that reduced lysoPCs essentially underlie the pathogenesis of AAA and that lysoPCs are promising biomarkers for AAA development.

## 1. Introduction

Abdominal aortic aneurysm (AAA) is hallmarked by irreversible dilation of the infrarenal aorta with a diameter above 3 cm, which is primarily asymptomatic as they grow, but the final rupture of the dilated aorta has a mortality rate over 90% [1]. Clinical AAA diagnosis is based on aortic imaging methods, such as ultrasound or computed tomography, and most aneurysms discovered by screening are of small size and do not need immediate surgical repair [2]. Nevertheless, the enlargement may be initially slow and then increases exponentially [3]. In this view, the identification of sensitive and specific biomarkers of growth and AAA rupture may identify metabolically active and potentially risky aneurysms, unveil novel pathological mechanisms and target new pharmacological inhibitions of growth. 

The typical clinical risk factors for AAA involve advanced age, male gender, smoking, hypertension, dyslipidemia and a family history of aneurysm disease [4]. Recent studies highlight the potential importance of lipid metabolism in AAA pathogenesis to a greater extent than previously thought. The largest genome-wide association study indicated the key role of aberrations of lipid metabolism in the development of AAA and risk alleles, including sortilin 1, low-density lipoprotein (LDL) receptor and phospholipid transfer protein regulating cholesterol transport [5]. Another human genetic association study further supports the importance of lipid metabolism in AAA pathogenesis, showing that plasma HDL- and LDL-cholesterol (HDL-c and LDL-c) levels are related to the likelihood of having an AAA diagnosed [6]. Most relevant studies are limited to measuring conventional lipid parameters with triglycerides and LDL-c positively related to AAA occurrence, and HDL-c negatively associated with AAA presence [6,7]. The main drawback is that this approach incorporates multiple lipids from different functional processes and cannot detect the subtle changes of specific lipids in their composition. Furthermore, clinical determinations of whole-lipid class abundance using enzymatic methods generally do not exhibit the content of individual lipid species in disease and health states. Thus, the specific role of dyslipidemia in AAA development is still under a veil.

Lipidomics, which emerged in 2003 [8], allow us to study lipid metabolism by quantifying the changes in individual lipids classes and their concentrations that reflect metabolic differences and seems to aid in the understanding of AAA pathogenesis when combining sensitive analytical techniques with multivariate analysis. Human body fluid, especially blood, contains a great source of information about individual differences in health, disease, diet and lifestyle. For metabolomics analysis of AAA, plasma or serum are most commonly selected due to their direct contact with the aorta and the simplicity of collecting and detecting them. Several studies that apply metabolomics and lipidomics to human AAA are carried out based on plasma/serum metabolite profiling [9,10,11]. Multiple lipids were identified between AAA and control subjects; however, these different lipids lack further verification and correlation analysis with clinical characteristics. 

In the present study, global lipid profiling of AAA patients and its association with clinical characteristics will be explored and the variation of distinct lipids found in a clinical study during the AAA formation will also be verified in a mouse model. This work will help gain a global metabolic insight into the pathological mechanism of AAA, and provide a reliable basis for disease detection, diagnosis and even treatment strategy.

## 2. Results

### 2.1. Baseline Characteristics of Enrolled Subjects

The clinical characteristics of the study participants are shown in Table 1. The difference in age, male-to-female ratio and BMI were not significant between controls and AAA. After adjustment for age, gender and BMI, subjects with AAA had a higher hypertension and CAD rate than those of control subjects (*p* < 0.05). Concentrations of HCY, UA, hsCRP and monocytes in subjects with AAA were significantly higher compared to controls (all *p* < 0.01). By contrast, blood levels of glucose, TC, HDL-c and platelets in patients with AAA were significantly lower than control subjects (*p* < 0.05) after adjusting for age, gender and BMI.

### 2.2. Changes in Plasma Lipidomics between Control and AAA Subjects

Enrolled participants included 70 diagnosed AAA patients and 36 age- and sex-matched non-AAA controls who underwent global lipidomic profiling by using the untargeted LC-MS/MS approach. Based on the VIP analysis and Student’s t-test, a total of 62 molecules in positive mode and 144 in negative mode were tentatively identified from the lipidomics data as differentially expressed metabolites (DEMs) with VIP > 1.00 and *p* < 0.05. OPLS-DA was conducted to visualize the distribution and the grouping of each sample (Figure 1A). The score plots showed that AAA and control subjects were well-separated. The top 20 lipids responsible for discrimination between AAA and control are presented in Table 2. Discrimination between AAA and controls can be principally attributed to lysophosphatidylcholines (LysoPCs) that were reduced in AAA. Then, we performed ROC analysis for the top five LysoPCs to determine their diagnostic performance, which ranged from 0.819 to 0.946 as shown in Figure 1B, and LysoPC (16:0) and lysoPC (18:0) had the highest scores of 0.946 and 0.939, respectively. A bubble plot was used to display the lipid content change, difference significance and classification information (Figure 1C). Each point represents a lipid, and the larger size of the point represents the smaller *p*-value from the Student’s t-test. These observations imply that the overall biosynthesis of lysoPCs may be preferentially reduced in AAA patients compared to controls. Furthermore, the identified molecules were subjected to MetaboAnalyst 5.0 for pathway enrichment. As shown in Figure 1D, there were multiple metabolic pathways perturbed in the disease state (only the top 8 most significantly enriched are listed), mainly including arachidonic acid metabolism, sphingolipid metabolism, glycerophospholipid metabolism as well as glycosylphosphatidylinositol (GPI)—anchor biosynthesis. In addition, the analysis showed that the plasma lysoPCs levels decrease with the AAA diameter (Figure 2A,B). For most lysoPCs, the area peak was the lowest in large AAA patients (lAAA, diameter > 50 mm), then in small AAA patients (sAAA, 30 mm < diameter < 50 mm), and the control group was the highest (Figure 2C–M). 

### 2.3. HDL-c and hsCRP Were Distinctly Associated with Altered Profiles of LysoPCs in AAA

In order to illustrate the interaction between lipids and clinical characteristics, correlations between serum lipids were analyzed using Spearman’s correlation for two groups. As shown in Figure 3, most strong correlations between lipids and clinical characteristics in control (Figure 3A) were weakened in AAA (Figure 3B), indicating a perturbation of lipid metabolism in AAA subjects. Negative associations between all plasma lysoPCs and CAD rate and hsCRP in controls were observed; however, the association became positive in AAA patients (Figure 3C,D). These data suggested that plasma lysoPCs may elicit specific physiological properties in inflammation for AAA progress. In contrast, the positive associations between most plasma lysoPCs and HDL-c, considered as “good cholesterol”, in controls disappeared in the AAA group, suggesting that decreased levels of plasma lysoPCs are harmful towards overall lipid metabolic health in AAA patients.

### 2.4. AAA Formation in ApoE ^−/−^ Mice and LysoPCs Decrease with AAA Progression

Male ApoE^−/−^ were randomly divided into two groups: the control group (n = 8, treated with saline) and the Ang II-induced AAA group (n = 8). Mice with aortic rupture and aortic dissection within the aorta upon aneurysm formation were excluded. Compared with those from mice treated with saline, the aortas of mice treated with Ang II showed a significant increase in aortic expansion after 4 weeks (Figure 4A). OPLS-DA model showed clear separations between the AAA group and the control group at 4 weeks from the score plot (Figure 4B). In the bubble plot, the relative difference of lysoPCs displayed a similar trend to human data, in which lysoPCs were obviously lower in AAA mice than control (Figure 4C). Metabolic pathway analysis indicated several pathways were altered, including GPI—anchor biosynthesis, sphingolipid metabolism and glycerophospholipid metabolism (Figure 4D), which were in line with the results from human plasma lipidomics (Figure 1D). In pursuit of revealing the lysoPCs alteration during the AAA progression, blood was collected at indicated time points 0, 2 and 4 weeks during the modeling. Lipidomic results demonstrate that lysoPCs involving LysoPC (20:0), LysoPC(O-18:0), LysoPC (18:1(9Z)), LysoPC (16:0) and LysoPC (15:0) increased in control mice but decreased in AAA mice during the modeling process (Figure 5A–F).

## 3. Discussion

Recent advancements in chromatography and mass spectrometry provide an excellent way to study pathophysiology and identify novel biomarkers for the diagnosis and development of AAA. In this work, untargeted lipidomics using UPLC-MS was applied to identify different lipidomes associated with AAAs and found LysoPCs were significantly reduced in AAA subjects and decreased with AAA development. 

As they have direct contact with the vascular endothelium, blood lipid species and concentrations not only affect the environment of the aorta, but are also affected by the metabolism of the aortic wall. Investigations revealed that serum LDL-c, TC and TG levels are positively correlated and HDL-c is negatively correlated with AAA [7]. Consistent with a previous study [12], we found that AAA had a higher hypertension and CAD rate, together with higher concentrations of HCY, UA, hsCRP and monocyte. Otherwise, glucose, TC, HDL-c and platelet in patients with AAA were significantly lower than those of control subjects. The lower TC in AAA is probably attributed to the small sample size or the possible use of cholesterol-reducing medications. Particularly, there are positive associations between most plasma lysoPCs and HDL-c in controls, and high levels of HDL-c are commonly considered cardioprotective. Whereas, this positive correlation disappears in the AAA group, suggesting that decreased levels of plasma lysoPCs are compounds with low levels of HDL-c in AAA, which indicates a possible interaction between lysoPCs and HDL-c in AAA development. Isolated HDLs from a subset of AAA patients or mice will help to better investigate HDL-specific lipid changes, especially the LysoPCs. On the other hand, CAD rate and CRP, a sign of inflammation in acute or chronic conditions, were both negatively related to all lysoPCs in controls. Then, the relationships all became positive in AAA indicating an underlying interaction between lysoPCs and coronary artery disease and inflammation. 

LysoPCs are a group of endogenous phospholipids considered an important cell-signaling molecule produced by the action of phospholipase A 2 (PLA_2_) on phosphatidylcholine. LysoPCs have been generally regarded as a group of potent pro-inflammatory and deleterious factors, which are linked to the development of atherosclerotic plaques [13] and endothelial cell dysfunction [14]. In addition, lysoPCs were increased in circulating modified LDL [15], enzymatically degraded LDL and oxidized LDL[16], which appears to be a major causative agent in the pathogenesis of atherosclerosis through the induction of inflammation and apoptosis, disruption of mitochondrial integrity, increase in oxidative stress and the inhibition of proliferation in endothelial cells [17]. Interestingly, as highlighted in recent studies, lysoPCs showed multiple beneficial properties under various pathological conditions. LysoPCs were found to be reduced in diabetes [18], pulmonary arterial hypertension [19], aging with impaired mitochondrial oxidative capacity [20], hepatocellular carcinoma tissues [21] and liver cirrhosis associated with increased mortality risk [22]. Consistently, lower levels of lysoPCs were also confirmed in 30 AAA patients compared to 11 healthy subjects with a trend related to the size of the aneurysm [9]. Another study observed increased secretion of lysoPCs by the aneurysm wall versus the health wall, as well as by the abluminal part of the intraluminal thrombus than by the luminal, which may partly explain the presence of a necrotic-lipid core in the aneurysm wall [10]. Moreover, a recent study demonstrated that levels of lysoPC (16: 0) and lysoPC (18: 2(9Z,12Z)) decreased with the development of AAA as well [11]. It was found that plasma lysoPCs exhibited a similar tendency as described in the previous study, which were declined in the AAA patients and its content was decreased with aneurysm enlargement. LysoPC (16:0) and lysoPC (18:0) were identified as the most abundant lysoPCs in plasma and the top two differentiating lipids with the lowest *p*-values (1.67 × 10^−18^ for lysoPC (16:0) and 1.28 × 10^−18^ for lysoPC (18:0)) across all 146 differentiating lipids. The declined levels of these two lysoPCs would largely contribute to the lower concentration of total lysoPCs in plasma. It is worth noting that LysoPC (16:0) and lysoPC (18:0) were also markedly reduced in the ruptured AAA (Appendix A) in relative to non-rupture AAA and non AAA controls. The results highlight the possibility to monitor these two lysoPC levels as the predictive signal for AAA rupture, which may greatly reduce the mortality of ruptured AAA. However, it needs larger cohorts and precise detection methods to confirm its potential role as a biomarker for AAA rupture. In addition, the blood sample was collected after AAA rupture; thus, whether the declined levels of these two lysoPCs is the cause or the result also needs further investigation. 

Of interest, arachidonic acid was elevated in both the plasma of AAA patients and mice, whereas the increase was not statistically significant in mice (Appendix A). Moreover, pathway enrichment analysis indicated arachidonic acid metabolism was perturbed in AAA patients. In another Danish study, gas chromatography examination with blood samples from 498 AAA patients showed arachidonic acid in AAA patients was also higher [23]. Meanwhile, declined linoleic acid in the AAA group was found, especially those with a large diameter (Appendix A). Arachidonic acid is synthesized from α-linolenic acid derived from linoleic acid and converted to eicosanoids [24]; it seems that the additional formation of arachidonic acid from linoleic acid can account for the lower levels of linoleic acid in AAA. Linoleic acid and arachidonic acid are major PUFAs comprising about 8% of all free fatty acids in the circulation [25]. Eicosanoids have multiple roles in inflammation [26], regulation of vasodilation, vascular permeability and recruitment of leukocytes, closely related to the pathogenesis of AAA. In addition, the increase in arachidonic acid is associated with the enhancement of the pro-inflammatory prostaglandin E 2 formation and inhibition of leukotriene B4 synthesis in AAA patients and mice plasma (Appendix A–I). LysoPCs were reported to stimulate the release of arachidonic acid in human endothelial cells [27] and coronary artery smooth muscle cells [28]. It seems that reduced levels of plasma lead to accumulated lysoPCs in the abluminal part of intraluminal thrombus, which can account for the enhanced release of arachidonic acid from artery endothelial cells to circulation in AAA. This statement needs to be further confirmed by combing the analysis of AAA thrombus composition, plasma lipids, endothelial and smooth muscle cell secretion. 

There are some limitations in this study. First, untargeted lipidomics was used to search for novel metabolic perturbations, and the identified differently expressed compounds just showed relative abundance, which was not precisely quantified and compared to established reference ranges. A more specific targeted lipidomic analysis is required to explore the concentrations of lysoPCs and to see if it can be widely used as a biomarker for AAA progress or even rupture. Secondly, only blood samples were studied in this work; tissue (e.g., AAA thrombus and aneurysm wall) and cells (e.g., endothelial and smooth muscle cells) from a well-designed AAA animal model or clinical samples will help better illustrate the pathophysiology of AAA. Moreover, PLA_2_ and downstream enzyme levels in circulation and lysoPC receptor expressions in aneurysm artery walls also play important roles in the regulation and function of lysoPCs, which deserves more investigation.

By means of untargeted lipidomics using UPLC-MS, a set of lysoPCs declined in both humans and mice with AAA, and the associations between lysoPCs and the clinical characteristics involving CAD rate, HDL-c and hsCRP were analyzed. Taken together, this study points out an essential role for lysoPCs in the development and progression of AAA, and lysoPCs, as a comparably stable compound, require an in-depth and intensive investigation to explore its role as a biomarker for AAA development.

## 4. Materials and Methods

### 4.1. Study Population

In this study, 70 patients with AAA and 36 control subjects without AAA were consecutively recruited between August 2020 and December 2021. Hypertension or diabetes was defined as a sitting blood pressure ≥140/90 mmHg [29] or a fasting blood glucose ≥ 7.0 mmol/L [30] as before [31]. AAA subjects were enrolled from patients undergoing computed tomographic angiography to assess AAA morphometry with an AAA diameter of >30 mm as part of their clinical diagnosis. The exclusion criteria included inflammatory AAA, dissecting AAA, false aneurysms, Takayasu arteritis and cancer. Non-AAA controls (AAA diameter <30 mm) were recruited from the physical examination center of PUMCH during the same period with matched age, sex, BMI, hypertension, dyslipidemia and diabetes. The exclusion criteria for controls involved infectious diseases, rheumatic immune diseases and cancer. The study protocol was approved by the Local Ethics Committee of the Department of Scientific Research at Peking Union Medical College Hospital (PUMCH) (Approved code: JS-2479 on 13 August 2020). Informed consent was obtained from all of the participants before entering the study.

### 4.2. Animal Model and Sample Collection

To establish AAA mouse models, ApoE^−/−^ mice (male) on the C57BL/6J background were obtained from Beijing Huafukang Bioscience. All mice were housed in a barrier facility and were provided with free access to food and water with an ambient temperature which ranged from 20 to 24 °C on a 12 h light–dark cycle. The model was generated by administering Ang II (1000 ng/kg per minute; Sigma-Aldrich, A9525, St. Louis, MO, USA) or saline to male mice (10 weeks of age) for 4 weeks via implantation of an Alzet mini-osmotic pump (Durect, Cupertino, CA, USA), as described in a previous study [32]. The diameter of the abdominal aortas of the mice was measured with a Vevo 2100 ultrasound system (Visual Sonics, Toronto, CA, USA) according to the protocol of the manufacturer, and an increase in the diameter by >50% was considered to indicate aneurysm formation. The mice blood was collected from tail vein at 0, 2 and 4 weeks during modeling process for lipidomic analysis. All animal studies were approved by the Institutional Animal Care and Use Committee of PUMCH, and the experiments conformed to the Guide for the Care and Use of Laboratory Animals.

### 4.3. Plasma Collection and Lipidomics Analysis

Human blood samples from an antecubital vein between 6:00 and 8:00 AM and venous blood draws from the mice tail were collected after 16 h of fasting. Fresh EDTA anticoagulated blood from humans and mice was centrifuged at 3000 g for 15 min at 4 °C. Plasma was decanted and stored in aliquots at −80 °C until analysis. High-resolution LC-MS/MS analysis was performed using a Thermo Q-Exactive mass spectrometer with a heated electrospray ionization source and a hybrid quadrupole-orbitrap mass analyzer. The C18 column (150 μm × 250 mm, 1.9 μm, Thermo, Sacramento, CA, USA) was used and the temperature was maintained at 45 °C. Mobile phase A consisted of H_2_O/ACN (6:4, *v*:*v*), 10 mM ammonium formate mixed with 0.1% formic acid and mobile phase B consisted of IPA/ACN (9:1, *v*:*v*), 10 mM ammonium formate mixed with 0.1% formic acid. The flow rate was 0.3 mL/min. The injection volume was 2 μL and the total analysis time lasted 20 min. Mass spectrometric detection was implemented on a Q-Exactive Orbitrap MS (Thermo, Sacramento, CA, USA) equipped in both positive and negative ion modes. The positive ion source voltage is 2.5 kV, and the negative ion source voltage is 4 kV. The capillary temperature was 320 °C. The liquid system was controlled using the Xcalibur 2.2 SP1.48 software, which also operated the data acquisition.

### 4.4. Statistical Analysis

For clinical parameters, statistical analyses used SPSS version 20.0 (Chicago, IL, USA). Normally distributed data were expressed as means ± SD. Data that were not normally distributed, as determined using the Kolmogorov–Smirnov test, were expressed as medians with an interquartile range. The Student’s t-test was used to evaluate differences between two study groups in normally distributed continuous variables. When normality was not confirmed, the Mann–Whitney U test was used. Three or more independent group comparisons were performed using a one-way analysis of variance (ANOVA). Metabolites for further statistical analysis were identified on the basis of a variable importance in projection (VIP) threshold of 1 from the orthogonal partial least square-discriminant analysis (OPLS-DA) model, which was validated at a univariate level with an adjusted *p* < 0.05. The molecules of interest were compared and identified using the HMDB database. OPLS-DA, heatmaps, metabolic pathway enrichment and classic univariate receiver operating characteristic (ROC) were achieved through MetaboAnalyst 5.0 software (http://www. metaboanalyst.ca/ (accessed on 10 May 2023)). ‘Wu Kong’ platform (https://www.omicsolution.com/wkomics/main/ (accessed on 10 May 2023)) was applied for relative correlation matrices of lipids analysis. Violin plots were achieved via bioinformatics platform (https://www.bioinformatic s.com.cn/ (accessed on 10 May 2023)).

## Figures and Tables

**Figure 1 ijms-24-10253-f001:**
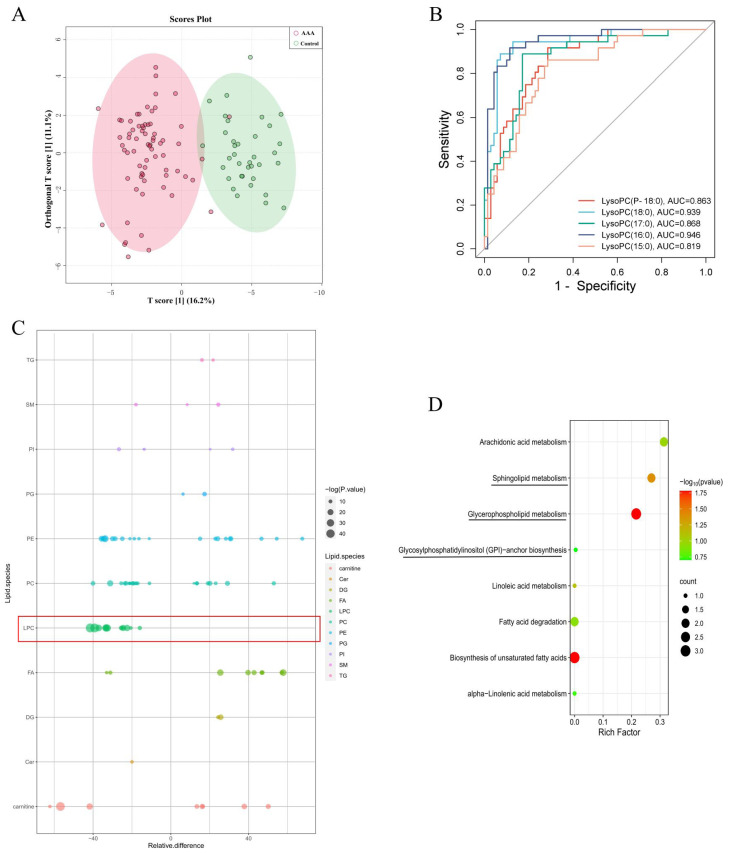
(**A**). OPLS-DA analysis among control group (Control, green circle), abdominal aortic aneurysm (AAA, red circle). (**B**). Classification performance based on classic univariate receiver operating characteristic (ROC). (**C**). Bubble plot for the lipid species in AAA and control groups. The ordinate indicates the lipid species, and the abscissa presents relative difference of lipids (relative difference = 100% ∗ (AAA-control)/control). (**D**). Pathway analysis of differentially expressed metabolites (DEMs). The ordinate indicates the pathway name, and the abscissa presents rich factor values of pathways (rich factor = number of DEMs enriched in the pathway/total number of all metabolites in the background).

**Figure 2 ijms-24-10253-f002:**
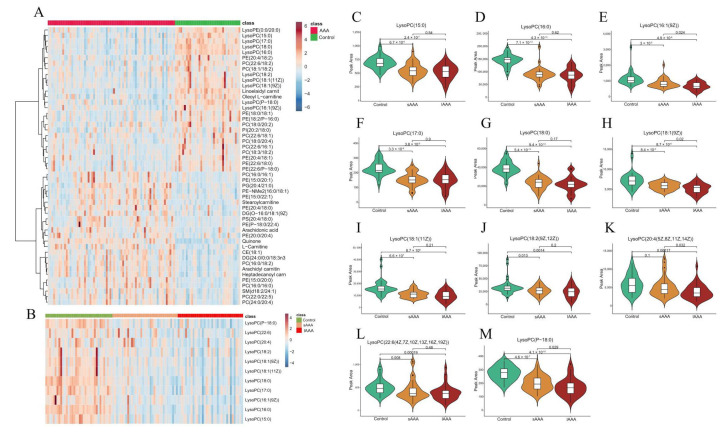
(**A**). Heatmap of hierarchical clustering of all lipids analysis for the AAA and control groups. (**B**). Heatmap of hierarchical clustering of lysoPCs analysis in control, sAAA and lAAA groups. Violin plots of relative abundance for the LysoPCs in control (n = 36), sAAA (n = 35) and lAAA (n = 35). The following figures displayed individual level of LysoPCs in three groups. (**C**). LysoPC (15:0), (**D**). LysoPC (16:0), (**E**). LysoPC(16:1(9Z)), (**F**). LysoPC(17:0), (**G**). LysoPC(18:0), (**H**). LysoPC(18:1(9Z)), (**I**). LysoPC (18:1(11Z)), (**J**). (18:2(9Z,12Z)), (**K**). LysoPC (20:4(5Z,8Z,11Z,14Z)), (**L**). LysoPC (22:6(4Z, 7Z, 10Z, 13Z, 16Z, 19Z)), (**M**). LysoPC(P-18:0).

**Figure 3 ijms-24-10253-f003:**
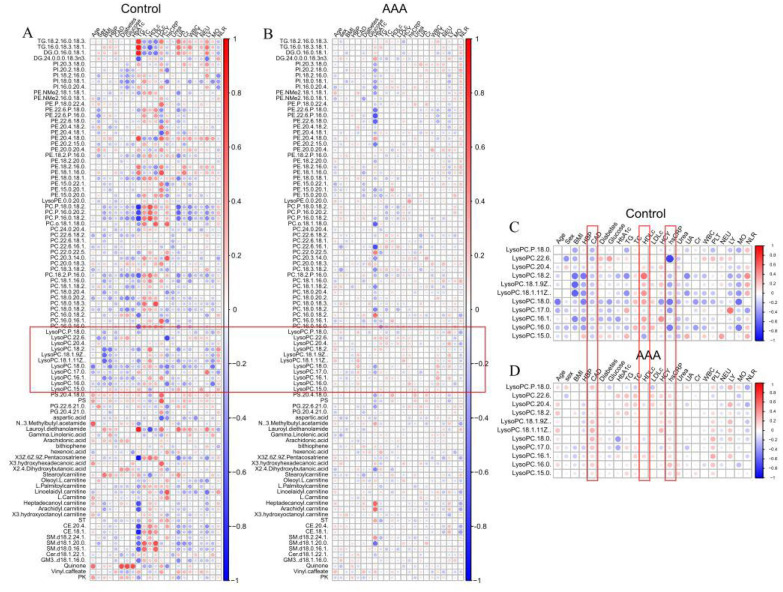
Spearman’s correlations between lipids and clinical characteristics were analyzed and presented. Correlations between plasma lipids and clinical characteristics were weakened in AAA patients (**B**) compared to non-AAA controls (**A**). Strengthened correlations were observed between plasma lysoPCs and multiple blood metabolic indices in non-AAA controls (**C**) compared to AAA subjects (**D**). Color of circles indicates direction of correlations (red: positive correlations; blue: negative correlations), and sizes and color intensities of circles indicate strengths of correlation (Bigger and darker circle indicates stronger correlation). Vertical scale indicates values of correlation coefficients.

**Figure 4 ijms-24-10253-f004:**
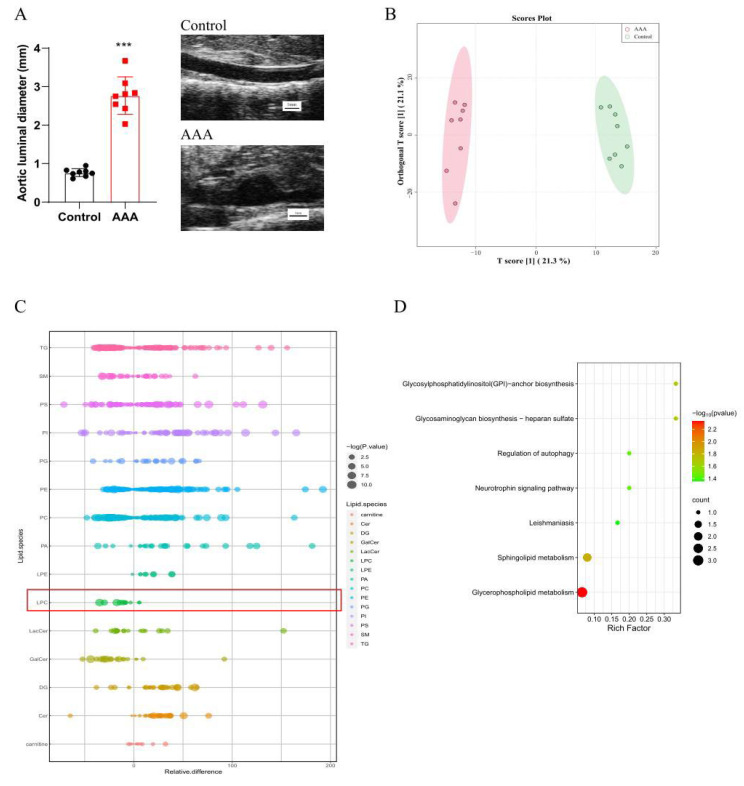
(**A**). Representative images to show high-frequency ultrasonography of abdominal aorta of AAA and control mice. The scale bar is 1 mm. (**B**). OPLS-DA analysis among control (green circle) and AAA (red circle) mice. (**C**). Bubble plot for the lipid species in AAA and control mice. (**D**). Pathway analysis of differentially expressed metabolites (DEMs). *** *p* < 0.001 vs. control mice.

**Figure 5 ijms-24-10253-f005:**
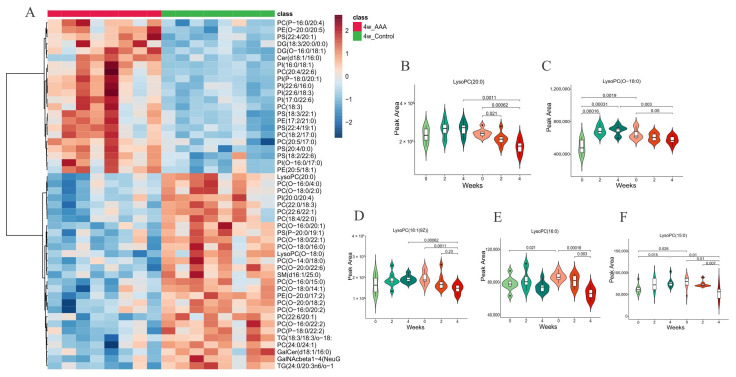
(**A**). Heatmap of hierarchical clustering analysis for the AAA and control mice. Violin plots of relative abundance changes of the LysoPCs at 0, 2 and 4 weeks in controls (n = 8) and AAA (n = 8) mice. (**B**). LysoPC(20:0), (**C**). LysoPC(O-18:0), (**D**). LysoPC(18:1(9Z)), (**E**). LysoPC(16:0) and (**F**). LysoPC(15:0).

**Table 1 ijms-24-10253-t001:** Summarizes the clinical characteristics of the recruited subjects.

Characteristics	Control (n = 36)	AAA (n = 70)	*p*-Value ^a^	*p*-Value ^b^
Age (years)	70.11 ± 8.67	70.37 ± 8.59	0.88	-
Gender (M/F)	33/3	61/9	0.75	-
BMI (Kg/m^2^)	25.3 ± 4.01	24.36 ± 3.49	0.214	-
Hypertension	17 (47.2%)	55 (78.6%)	*** (0.002)	*** (0.000)
Diabetes	12 (33.3%)	11 (15.7%)	* (0.048)	0.057
CAD	1 (0.03%)	20 (28.6%)	*** (0.0015)	*** (0.001)
Glucose (mmol/L)	5.60 (1.68)	5.00 (1.15)	*** (0.002)	* (0.021)
HbA1c (%)	5.7 (1.05)	5.85 (1.48)	-	-
HCY (μmol/L)	13.5 (2.8)	15.2 (7.7)	* (0.029)	* (0.016)
TC (mmol/L)	4.76 ± 0.76	4.37 ± 1.28	-	* (0.023)
TG (mmol/L)	1.26 (1.24)	1.21 (1.29)	-	-
HDL-c (mmol/L)	1.14 ± 0.29	0.98 ± 0.24	*** (0.004)	*** (0.000)
LDL-c (mmol/L)	2.91 ± 0.79	2.79 ± 1.00	-	-
hsCRP (mg/L)	1.05 (0.52)	1.85 (3.64)	* (0.022)	* (0.048)
Urea (mmol/L)	5.48 (2.09)	6.32 (3.26)	-	-
UA (μmol/L)	322.03 ± 69.30	368.27 ± 76.79	** (0.004)	*** (0.000)
Cr(E) (μmol/L)	83 (20.75)	85 (31)	-	-
WBC (10^9^/L)	6.08 (2.12)	6.38 (2.33)	NS	NS
Platelet (10^9^/L)	213.03 ± 59.05	185.4 ± 64.2	* (0.034)	* (0.029)
Monocyte (10^9^/L)	0.355 (0.13)	0.39 (0.22)	* (0.03)	** (0.012)
Neutrophil (10^9^/L)	3.82 (1.68)	3.89 (1.81)	-	-
Lymphocyte (10^9^/L)	1.78 ± 0.60	1.56 ± 0.56	-	-
NLR	2.09 (1.21)	2.47 (1.66)	-	-

Data are presented as mean values and SD, counts, or medians and interquartile ranges, as appropriate. CAD, Coronary artery disease; HCY, Homocysteine; TC, Total cholesterol; TG, Triglycerides; HDL, High-density lipoprotein; LDL, Low-density lipoprotein; FFA, Free fatty acid; hs-CRP, high-sensitivity C-reactive protein; TBA, Total bile acid; UA, Uric acid; Cr(E), Creatinine; WBC, White blood cell; NLR, Neutrophil/Lymphocyte. ^a^ *p*-value, AAA vs. control. ^b^ *p*-value, after adjusting for age, gender and BMI. * *p* < 0.05, ** *p* < 0.01, *** *p* < 0.001 vs. control subjects.

**Table 2 ijms-24-10253-t002:** Table of top 20 lipids ranked by descending VIP scores from OPLS-DA analysis classifying AAA and control.

Rank	Lipids	VIP Score	Fold Change
1	LysoPC(16:0)	1.955664231	0.60829
2	LysoPC(18:0)	1.938433674	0.58536
3	Linoelaidyl carnitine	1.933833948	0.43217
4	LysoPC(17:0)	1.679655063	0.66623
5	Quinone	1.656635156	1.6062
6	LysoPC(15:0)	1.597648917	0.77493
7	LysoPE(0:0/20:0)	1.581730908	0.66221
8	LysoPC(P-18:0)	1.535502986	0.67357
9	Oleoyl L-carnitine	1.395659551	0.58249
10	LysoPC(18:1(11Z))	1.364155556	0.62939
11	Arachidyl carnitine	1.34257126	1.1618
12	DG(24:0/0:0/18:3n3)	1.341787518	1.2566
13	PC(22:6(4Z,7Z,10Z,13Z,16Z,19Z)/18:2(9Z,12Z))	1.323787935	0.68801
14	LysoPC(18:1(9Z))	1.323317442	0.75435
15	CE(18:1(11Z))	1.317468625	1.4777
16	LysoPC(16:1(9Z))	1.287512004	0.66547
17	PC(16:0/18:2(9Z,12Z))	1.23655766	1.2012
18	Heptadecanoyl carnitine	1.221290467	1.134
19	L-Carnitine	1.219465979	1.3779
20	PE(18:0/18:1(9Z))	1.172025083	0.64375

PC, Phosphatidylcholine; DG, Diacylglycerol; CE, Cholesterol ester; PE, Phosphatidylethanolamine.

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
