# Peer review of "Plasma Lipidomics Analysis Reveals the Potential Role of Lysophosphatidylcholines in Abdominal Aortic Aneurysm Progression and Formation"

_ijms, 2023, doi:10.3390/ijms241210253_

Round 1

Reviewer 1 Report

This manuscript describes the influence of lipids, especially lysoPC, on the formation of abdominal aortic aneurysm. I have some comments to the author.

Comment 1. Intended for persons undergoing contrast-enhanced CT, the background of subjects in the AAA and non-AAA groups should be shown. Age, male-to-female ratio, BMI, blood pressure, etc. are described in the Results section. However, no specific numerical values are given, and the content is extremely insufficient. Is it desirable to show the background of the subjects in both groups in a table or the like? Of course, the background such as underlying diseases should also be shown.

Comment 2. The method does not describe the collection of human samples, what was measured, etc. In addition, Insufficient description of mouse sample processing up to LC/LM/MS.

Comment 3. There is a statement in the Results that “The top twenty lipids responsible for discrimination between AAA and control were presented in Table 2.”, but there is no ”Table” in this manuscript. Even though "Table 1" is not shown in the first place, "Table 2" suddenly appears!

Comment 4. Results bridges the results of Fig 2C, but the figures, including "Fig 2C-M", have low resolution, so the numbers cannot be read.

Comment 5. Based on the results of this experiment and past reports, the importance of lysoPC is described, and the discussion also mentions that arachidonic acid was elevated in the plasma of AAA patients and AAA mice. leukotriene, prostaglandin, etc.)?

Author Response

Thanks very much for taking your time to review the manuscript. We sincerely appreciated all your suggestions and comments. Please find the itemized responses in below and the ‘Track changes’ feature in Microsoft Word and all changes made are easily identifiable in our paper.

Reviewer1

This manuscript describes the influence of lipids, especially lysoPC, on the formation of abdominal aortic aneurysm. I have some comments to the author.

Comment 1. Intended for persons undergoing contrast-enhanced CT, the background of subjects in the AAA and non-AAA groups should be shown. Age, male-to-female ratio, BMI, blood pressure, etc. are described in the Results section. However, no specific numerical values are given, and the content is extremely insufficient. Is it desirable to show the background of the subjects in both groups in a table or the like? Of course, the background such as underlying diseases should also be shown.

Response:

Thanks for your advice. Yes, the background of the participants involving the specific numerical values were listed in table1 which may be missed in the first submission for some reasons and have been supplied now for your reference.

Comment 2. The method does not describe the collection of human samples, what was measured, etc. In addition, Insufficient description of mouse sample processing up to LC/LM/MS.

Response:

Thanks for the good question. Human blood samples from an antecubital vein between 6:00 and 8:00 AM and venous blood draws from the mice tail were collected after 16 hours of fasting. Fresh EDTA anticoagulated blood from human and mice was centrifuged at 3000g for 15 min at 4℃. Plasma was decanted and stored in aliquots at -80℃ until analysis. Both human and mice plasma were analyzed by the High-resolution LC-MS/MS as described in the method.

PS: The clinical characteristics including BMI, hypertension, diabetes, Coronary artery disease, Glucose, HbA1c, HCY, TC, TG, HDL-c, LDL-c, FFA, hs-CRP, TBA, UA, Cr(E), WBC, NLR were collected from clinical measurement.

Comment 3. There is a statement in the Results that “The top twenty lipids responsible for discrimination between AAA and control were presented in Table 2.”, but there is no ”Table” in this manuscript. Even though "Table 1" is not shown in the first place, "Table 2" suddenly appears!

Response:

Thanks a lot for your kind remind. The table 1 and 2 have been added and the relevant locations are “The clinical characteristics of the study participants are shown in Table 1.” and “The top twenty lipids responsible for discrimination between AAA and control were presented in Table 2.”

Comment 4. Results bridges the results of Fig 2C, but the figures, including "Fig 2C-M", have low resolution, so the numbers cannot be read.

Response:

Thanks for your comment, and the numbers in Fig 2C-M is indeed too small to read. We have modified the Fig 2C-M and Fig 5B-F to make it large enough for reading.

Comment 5. Based on the results of this experiment and past reports, the importance of lyso/PC is described, and the discussion also mentions that arachidonic acid was elevated in the plasma of AAA patients and AAA mice. leukotriene, prostaglandin, etc.)?

Response:

Esterified arachidonic acid (AA) on the inner surface of the cell membrane is hydrolyzed to its free form by phospholipase A2, which is in turn further metabolized by cyclooxygenases and lipoxygenases and cytochrome P450 enzymes to a spectrum of bioactive mediators that includes prostaglandins (PGs), leukotrienes (LTs), epoxyeicosatrienoic acids, eicosatetraenoic acids, and so on. AA has already been reported higher in AAA patients in several studies (see the discussion), and individuals in the upper tertile of arachidonic acid had higher probability of having AAA [1]. AA metabolism is crucial to the inflammatory process and its resolution. It is believed that AA is harmful, because its administration enhances PGE2 formation, a pro-inflammatory molecule. PGE2 inhibits LTB4 synthesis by modulating the expression of lipoxygenases [2].

In this study, PGE2 was observed increased in AAA patients and mice at 4 weeks (Supplementary figure G and I). The levels of PGE2 was also proved significantly higher in ruptured AAAs than in non-ruptured AAAs of similar diameter [3]. Another research found the down-regulation of PGE2 levels result in the amelioration of inflammation [4]. Previous study suggested that human AAA wall tissue converts arachidonic acid and the unstable epoxide LTA4 into significant amounts of cysteinyl-leukotrienes (LTC4, LTD4 and, LTE4) and to a lesser extent dihydroxy LTB4 [5]. Based on this conclusion, we further analyze the data. Unfortunately, the UPLC-MS did not identify any cysteinyl-leukotrienes both in human and mice samples. We only found LTB4 was indeed reduced in AAA human and mice group (Supplementary figure F and H).

In summary, the increase of arachidonic acid is associated with the inflammation in AAA patients, which enhances the pro-inflammatory PGE2 formation, and LTB4 synthesis is inhibited in AAA condition. This content has been added in the discussion.

Reviewer 2 Report

1, there are several methods to evaluate the size of AAA, which method did the authors use?

2, Would the statin therapy influence the result? should be discussed

3, Local factors also contributed to AAA, the authors should examine the local changes in mice model to confirm their hypothesis

acceptable 

Author Response

Thanks very much for taking your time to review the manuscript. We sincerely appreciated all your suggestions and comments. Please find the itemized responses in below and the ‘Track changes’ feature in Microsoft Word and all changes made are easily identifiable in our paper.

Reviewer 2

1, there are several methods to evaluate the size of AAA, which method did the authors use?

Response:

According to the 2022 ACC/AHA Guideline for the Diagnosis and Management of Aortic Disease, CTA, MRI, Vascular ultrasound, Echocardiography, and Intravascular Ultrasound were mainly applied for measurements of the aorta to characterize aortic disease and guide treatment decisions.

Computed Tomography Angiography (CTA) can image the entire aorta and its branches with high spatial resolution and fast acquisition, which has a very high sensitivity and specificity for acute aortic syndromes and traumatic aortic injuries. Thus, CTA was used to evaluate the size of AAA in this research.

2, Would the statin therapy influence the result? should be discussed

Response:

Thanks for the valuable comment, the plasma from AAA patients was collected at the first morning after their admission. The statin has not been administered at the blood collection if the patient is diagnosed as hyperlipidemia. The non AAA control was also selected without the statin history.

3, Local factors also contributed to AAA, the authors should examine the local changes in mice model to confirm their hypothesis

Response:

Yes, the local changes of aortic wall from mice will better confirm the hypothesis and is the focus of our future study. In the present research, reduced levels of plasma lysoPCs may due to accumulated lysoPCs in the abluminal part of intraluminal thrombus, which can also account for the enhanced release of arachidonic acid from artery endothelial cells to circulation in AAA. This statement needs to be confirmed by combing analysis of AAA thrombus composition, plasma lipids, endothelial and smooth muscle cells secretion in mice model. In particular, tissue (e.g. AAA thrombus, aneurysm wall) and primary cells (e.g. endothelial and smooth muscle cells) from a well-designed AAA mice model or clinical samples will help better illustrate the pathophysiology of AAA.

Round 2

Reviewer 1 Report

Thank you for the revised manuscript. 

I don't think the Figures have been improved as they are still very detailed and hard to see, but I wondered if it could be helped because there is a lot of data.

Reviewer 2 Report

the authors has answered the reviewer's question